# Impact of the COVID-19 Pandemic on Microbial Profiles and Clinical Outcomes in Orbital and Preseptal Cellulitis

**DOI:** 10.3390/microorganisms12112262

**Published:** 2024-11-08

**Authors:** Yu-Ting Tsao, Yueh-Ju Tsai, Chau-Yin Chen, Yen-Chang Chu, Yun-Shan Tsai, Yi-Lin Liao

**Affiliations:** 1Department of Ophthalmology, Chang Gung Memorial Hospital, Linkou Medical Center, Taoyuan 333423, Taiwan; sherry450047@gmail.com (Y.-T.T.); erinyjtsai@gmail.com (Y.-J.T.); umbo2003@gmail.com (Y.-C.C.); 2Department of Medicine, Chang Gung University College of Medicine, Taoyuan 333323, Taiwan; m7043@adm.cgmh.org.tw; 3Department of Ophthalmology, Chang Gung Memorial Hospital, Chiayi 61363, Taiwan; 4School of Traditional Chinese Medicine, Chang Gung University College of Medicine, Taoyuan 333323, Taiwan; cathytsai1207@gmail.com

**Keywords:** pandemic, COVID-19, SARS-CoV-2, orbital cellulitis, preseptal cellulitis, microbiology

## Abstract

Orbital cellulitis and severe preseptal cellulitis are critical periocular infections with potential vision- and life-threatening implications. The COVID-19 pandemic is hypothesized to have had an influence on their presentation and pathogenesis; however, the real impact remains unclear. In this retrospective multicenter cohort study from January 2017 to December 2022, we analyzed 1285 cases with preseptal or orbital cellulitis in pre-pandemic (2017–2019) and pandemic (2020–2022) cohorts. A notable decrease in hospitalized cases during the pandemic period was observed (97 patients in the pre-pandemic group vs. 54 in the pandemic group, *p* = 0.004), particularly among individuals aged 30–39 (*p* = 0.028). Sinusitis remained the leading cause, but odontogenic cases increased (*p* = 0.025). In addition, microbial diversity decreased during the pandemic, with the effective number of species decreasing from 17.07 to 8.87, accompanied by a rise in antibiotic resistance, notably against erythromycin, oxacillin, penicillin, and metronidazole. While visual outcomes appeared worse in the pandemic group, statistical significance was not reached. These findings suggest that the characteristics, etiology, microbial profiles, resistance patterns, and visual outcomes of orbital and preseptal cellulitis have undergone alterations post-COVID-19 pandemic. Vigilance in clinical management and public health measures is crucial, with further research needed to optimize treatment strategies.

## 1. Introduction

Orbital cellulitis is an emergent medical condition carrying the potential for severe sight- and life-threatening sequelae. It is characterized by infection of the soft tissue posterior to the orbital septum, distinguishing it from preseptal cellulitis, which affects the tissue in front of this barrier. Notably, preseptal cellulitis can progress to orbital cellulitis and induce serious complications. These complications include orbital compartment syndrome, retinal artery or vein occlusion, encephalomeningitis, brain abscess, and cavernous sinus thrombosis [1,2,3]. Due to the risk of these serious complications, understanding the symptoms, signs, prevalence, etiology, microbiology, and antibiotic efficacy of both orbital and preseptal cellulitis is crucial for ensuring appropriate treatment and minimizing the risk of adverse outcomes. While numerous studies have extensively reported the age distribution, seasonal variation, pathogens, and antibiotic efficacy associated with preseptal and orbital cellulitis [4,5,6], it is worth noting that there are currently no investigations into the influence of the COVID-19 pandemic on these factors.

COVID-19, caused by the SARS-CoV-2 virus, is a respiratory illness that emerged in late 2019 and spread rapidly worldwide, leading to a global pandemic. Existing research on COVID-19 and ocular health predominantly focuses on topics such as case reports of COVID-19-related orbital cellulitis, transmission of COVID-19 through ocular routes, ocular surface infections by COVID-19, and changes in eye care systems in response to the pandemic [7,8,9,10,11]. However, there remains a lack of studies addressing the impact of the COVID-19 pandemic on ocular microbiology and its implications for preseptal and orbital cellulitis. The COVID-19 pandemic has affected humanity not only through direct infection but also through profound changes in human lifestyle behaviors, encompassing widespread mask-wearing, frequent use of detergents and household cleaners, shifts in medical consultation patterns, delays in seeking medical care, and vaccination campaigns [12,13]. These behavioral shifts could exert an influence on the microbiome and environmental microbiology, potentially resulting in altered characteristics of infectious diseases. Consequently, investigating changes in the characteristics, etiologies, isolated pathogens, and antibiotic efficacy following the COVID-19 pandemic could inform the development of new prevention and treatment strategies.

In this study, our objective is to examine the impact of the COVID-19 pandemic on microbial profiles and clinical outcomes in orbital and preseptal cellulitis. By gaining a better understanding of this sight- and life-threatening condition in the context of the COVID-19 era, we aim to improve management and outcomes for affected individuals.

## 2. Materials and Methods

### 2.1. Participants

This retrospective multicenter cohort study encompasses clinical data gathered from patients admitted to multiple branches of Chang Gung Memorial Hospital, namely Taipei, Linkou, Taoyuan, Keelung, and Chiayi, with confirmed diagnoses of preseptal cellulitis or orbital cellulitis between January 2017 and December 2022. This study was conducted in strict accordance with the ethical principles delineated in the Declaration of Helsinki and received explicit approval from the institutional review board of Chang Gung Memorial Hospital (IRB numbers: 202201723B0). Notably, Taiwan recorded its initial case of SARS-CoV-2 infection in January 2020 [14]. Consequently, patients were categorized into two groups based on the timing of admission: the pre-pandemic cohort (2017–2019) and the pandemic cohort (2020–2022). Inclusion criteria comprised patients assigned International Classification of Diseases, Ninth Revision (ICD-9) codes including 376.0 (acute inflammation of orbit), 376.00 (acute inflammation of orbit, unspecified), and 376.01 (orbital cellulitis), as well as those assigned ICD-10 codes including H05.00 (unspecified acute inflammation of orbit), H05.01 (cellulitis of orbit), H05.011 (cellulitis of right orbit), H05.012 (cellulitis of left orbit), H05.013 (cellulitis of bilateral orbits), and H05.019 (cellulitis of the unspecified orbit). Patients with mild preseptal cellulitis or hordeolum without admission were excluded from this study. Additionally, individuals with preseptal and orbital cellulitis other than bacteria origin or aseptic orbital inflammation were excluded. We systematically reviewed the medical records of all patients to confirm their eligibility for inclusion in this study. The recruitment process is visually depicted in Figure 1A, elucidating the sequential flow of subjects throughout this study.

### 2.2. Data Collection

We collected patient data through medical chart review, including age, gender, BMI, disease diagnosis, laterality of orbital and preseptal cellulitis, underlying health conditions, presenting symptoms and signs, duration of hospitalization, microbiological culture findings, and treatment regimens. Surgical interventions were administered based on clinical necessity. Pathogen cultures were categorized into blood, localized noninvasive (including conjunctival swabs and wound pus culture), and surgical cultures obtained during drainage procedures, with surgical cultures serving as definitive pathogen identification. Antimicrobial susceptibility of isolated pathogens was routinely evaluated using the disk diffusion method in accordance with the Clinical and Laboratory Standards Institute guidelines [15]. Visual acuity (VA) and intraocular pressure (IOP) were documented initially and 3 months after discharge. VA values were transformed into logMAR units for statistical analysis, with specific conversions applied for semiquantitative measurements: LogMAR of 1.85 for counting fingers vision, 2.3 for hand movement vision, 2.8 for light perception, and 3.0 for no light perception [16,17]. IOP measurements were performed using an iCare tonometer (Tiolat Oy, Helsinki, Finland).

### 2.3. Statistical Analyses

We used Cochran–Armitage analysis to assess trends in hospitalized cases of preseptal and orbital cellulitis. Descriptive statistics were employed to characterize patient demographics, with means and standard deviations or proportions presented as appropriate. The normality of continuous data distribution was assessed using the Kolmogorov–Smirnov test. Student’s *t*-test was applied for normally distributed data, while the Mann–Whitney U test was utilized for non-normally distributed data when comparing values between two groups. Proportional parameters between groups were compared using either the Chi-square test or Fisher’s exact test, depending on frequency assumptions. We used the Shannon H′ diversity index with an effective number of species to describe the microbiome diversity. Statistical analyses were conducted using IBM SPSS Statistics for Windows, Version 22.0 (IBM Corp., Armonk, NY, USA). A significance level of *p* < 0.05 was set to determine statistical significance.

## 3. Results

### 3.1. Participants and Characteristics of the Study Cohort

Between 2003 and 2022, a total of 3810 patients were diagnosed with orbital inflammation, identified through ICD codes including 376.0, 376.00, and 376.01 (ICD-9) or H05.00, H05.01, H05.011, H05.012, H05.013, and H05.019 (ICD-10). The overall admission rate during this period was 27.11%. While a downward trend was observed, it did not achieve statistical significance (*p* > 0.05). Notably, following the COVID-19 pandemic (2020–2022), there was a substantial decrease in admission rates compared to the pre-pandemic period (2017–2019) (16.2% vs. 24.0%, *p* = 0.004) (Figure 1B). After excluding inappropriate cases, further analysis was conducted on 97 admitted patients from the pre-pandemic group and 54 from the pandemic group. Table 1 provides an overview of the demographics and clinical characteristics of the participants. On average, patients in the pre-pandemic group were hospitalized for 11.54 ± 15.05 days, whereas those in the pandemic group stayed for 12.02 ± 17.59 days. The proportion of patients admitted due to pre-septal cellulitis significantly increased in the pandemic period (29.6% compared to 15.5%, *p* = 0.039). While average age did not exhibit a significant difference, upon stratifying age into intervals of 10 years from 0 to 99, it was observed that patients aged 0–9 showed the highest admission rate in both groups. However, patients aged 30–39 exhibited a significant decline after the COVID-19 pandemic (*p* = 0.028) (Figure 2A). Furthermore, seasonal distribution indicated an increased incidence during the summer in both groups (Figure 2B).

### 3.2. Predisposing Factors and Culture Results

Table 2 illustrates the predisposing factors associated with preseptal and orbital cellulitis. In the pre-pandemic group, sinusitis, skin infection, and hordeolum emerged as the most prevalent predisposing factors, whereas in the pandemic group, sinusitis, skin infection, and odontogenic factors were predominant. Notably, the percentage of patients with odontogenic etiology was significantly higher in the pandemic group (11.1% compared to 2.1%, *p* = 0.025).

Localized noninvasive cultures were obtained from 40 patients in the pre-pandemic group, revealing positive results in 32 cases, with a total of 23 distinct aerobic bacterial genera and 11 anaerobic bacterial genera identified. Polymicrobial findings were observed in 13 patients (40.6%). Similarly, in the pandemic group, cultures were obtained from 31 patients, yielding positive results in 26 cases, with a total of 16 different aerobic bacterial genera and 8 anaerobic bacterial genera identified. Polymicrobial results were noted in 12 patients (46%). *Staphylococcus aureus* emerged as the most frequently isolated pathogen in both periods. However, no significant differences were observed in the prevalence of any bacterial genera between the two groups. Further details of the microbiological findings are provided in Figure 3 and Appendix A.

Surgical drainage procedures were performed in 12 cases within the pre-pandemic group and 10 cases within the pandemic group. All surgical cultures yielded positive results, revealing a total of 22 different bacterial genera (H′ index = 2.84, effective number of genera = 17.07) in the pre-pandemic group and 12 different genera (H′ index = 2.18, effective number of genera = 8.87) in the pandemic group. Polymicrobial findings were observed in nine patients (75%) in the pre-pandemic group and four patients (40%) in the pandemic group, though this difference was not statistically significant (*p* = 0.192). Further details on the microbiological outcomes can be found in Figure 4 and Appendix A. In the pre-pandemic group, six patients underwent both localized noninvasive cultures and surgical drainage culture procedures. Among these, four patients exhibited at least one identical pathogen between the two types of cultures, while the remaining two patients displayed inconsistent pathogen profiles between noninvasive cultures and surgical drainage cultures. On the other hand, in the pandemic group, three patients underwent both localized noninvasive cultures and surgical drainage cultures, with all patients demonstrating consistent pathological reports.

### 3.3. Antibiotic Use, Drug Susceptibility, and Treatment Outcome

All patients received intravenous antibiotic therapy upon admission, with the most common regimen being intravenous amoxicillin–clavulanate, followed by ceftriaxone and vancomycin in both groups (Appendix A). The sensitivity rates of these three antibiotics against pathogens identified in localized noninvasive cultures were 60%, 80%, and 100% in the pre-pandemic group and not applicable (N/A), 100%, and 100% in the post-pandemic group. Furthermore, the sensitivity rates of cefazolin, oxacillin, and penicillin in the pre-pandemic group were all below 50%, whereas in the pandemic group, only the sensitivity rate of penicillin fell below 50%. Additional details are provided in Table 3. Regarding pathogens identified in surgical drainage cultures, the sensitivity rates of erythromycin and penicillin in the pre-pandemic group were below 50%, while in the pandemic group, the sensitivity rates of erythromycin, oxacillin, penicillin, and metronidazole were all below 50%. Notably, the lowest sensitivity rate was observed for penicillin (8.3%) in the pandemic group. Further details regarding the drug susceptibility of pathogens identified in surgical drainage cultures are provided in Table 4.

Following treatment, complete records of pre- and post-treatment visual examinations were available for 26 patients in the pre-pandemic cohort and 14 patients in the pandemic cohort. No significant differences were observed in terms of visual acuity or intraocular pressure outcomes between these two groups. While the pandemic group exhibited higher LogMAR visual acuity (indicating worse vision) at both admission and last follow-up, and exhibited less improvement following treatment, these differences did not achieve statistical significance (*p* > 0.05). Details of the VA outcomes are illustrated in Table 5.

## 4. Discussion

This is a pioneering study analyzing the epidemiological, prevalence, etiological factors, microbial profile and resistance, treatment, and prognosis changes in orbital cellulitis and preseptal cellulitis after the COVID-19 outbreak. We observed a significant decrease in the number of hospitalized patients with both orbital cellulitis and preseptal cellulitis during the COVID-19 pandemic, with a notable increase in the proportion of hospitalized patients with preseptal cellulitis. Further analysis revealed that the difference in numbers primarily occurred from the population aged 30–39 years, irrespective of seasonality, as summer remained the peak season for hospitalization in both pre- and post-COVID-19 pandemic groups. Regarding predisposing factors, sinusitis emerged as the most common predisposing factor in both pre- and post-COVID-19 pandemic groups. However, during the pandemic, there was a significant decrease in the proportion of sinusitis cases and an increase in cases with odontogenic etiology. This shift may be associated with mask-wearing practices and reduced routine dental visits, and our findings are consistent with those of previous studies [18,19,20]. In terms of microbial profile, we noted a decreasing trend in genera diversity during the pandemic period, with the effective number of genera decreasing from 17.07 to 8.87 species. Concurrently, there was an increasing trend in antibiotic resistance according to the surgical drainage outcome, particularly against erythromycin, oxacillin, penicillin, and metronidazole. However, these trends did not reach statistical significance. Finally, concerning prognosis, pandemic patients exhibited poorer performance in terms of admission visual acuity, visual acuity at the last follow-up, and the degree of improvement in visual acuity, although these differences did not reach statistical significance.

We propose that the substantial decrease in hospital admissions during the COVID-19 pandemic may be attributed to multiple factors. Firstly, there was a notable decrease in emergency department and outpatient visits, possibly due to public health interventions and concerns about viral exposure [21,22]. Concurrently, there was a significant increase in the utility of over-the-counter antibiotics, which could influence this trend [23,24]. However, it is important to recognize that excessive use of over-the-counter antibiotics can contribute to the development of resistant pathogens [22] and may reduce microbial species diversity [25].

Recent studies on gut, respiratory, oral, and plasma microbiome have highlighted these findings, such as dysbiosis, reduced species diversity, and an increase in pathogenic bacteria after the COVID-19 pandemic, which could exacerbate the outcomes of infectious disease [26,27,28,29,30,31]. In addition, long-term consequences of COVID-19 infection, according to those observed during the SARS epidemic, may heighten susceptibility to bacterial infections [32]. Therefore, it is imperative to reassess the prevalence and characteristics of infectious diseases, particularly those associated with significant mortality and morbidity, such as orbital cellulitis.

Continuously updating microbiological profiles is crucial for selecting the most effective treatments for ocular infections. Our study aligns with previous research conducted over the past 10 to 20 years, which consistently identified *Staphylococcus aureus*, coagulase-negative staphylococci, and Streptococcus spp. as the predominant pathogens [4,33,34]. Of particular concern is the rising prevalence of methicillin-resistant *Staphylococcus aureus* (MRSA), a significant public health threat [35]. Our study found a similar prevalence of MRSA in both pre-pandemic (66%) and pandemic groups (60%), consistent with previous reports ranging from 51.4% to 67% [4,34]. However, in cases where surgical drainage and culture confirmed true pathogenicity, the pandemic group had three cases of MRSA (100% of *Staphylococcus aureus* identified), whereas the pre-pandemic group had none (0% of *Staphylococcus aureus* identified). Despite the small sample size, this trend may indicate a potentially rapid increase in resistant *Staphylococcus aureus*. Additionally, regarding species diversity, we calculated the H’ diversity index in previous studies, ranging from 2.46 to 2.71, with an effective number of species between 11.67 and 14.99 [4,34]. However, in our pandemic cohort, there was a significant decrease in species diversity. This disparity may be attributed to shifts in human lifestyle and environmental microbiology resulting from the COVID-19 pandemic [36,37]. Nevertheless, further investigation and longitudinal studies are required to confirm this observed change.

There are few studies that offer comprehensive data on prevalence, etiology, microbial profiles, and resistance for orbital and preseptal cellulitis. Additionally, our study evaluates various outcome measures, such as visual outcomes and hospitalization duration. Furthermore, we provide innovative insights by investigating the impact of COVID-19 on these visually threatening infectious diseases. While our findings offer valuable insights, this study is subject to certain limitations. Firstly, it is an observational study conducted over a relatively short period, necessitating further long-term research to fully assess the impact of the COVID-19 pandemic on preseptal and orbital cellulitis. Secondly, while wound cultures or conjunctival swabs are prone to contamination, surgical cultures provide a more reliable pathogen report. However, the low number of patients undergoing drainage surgery limited our ability to thoroughly identify the true pathogen. Third, our statistical method using the Shannon H’ diversity index and the effective number of species has limitations, including sensitivity to rare species, challenges in interpretation, and assumptions of uniform sampling effort and a closed system. Fourth, not all patients in this study underwent complete cultures and visual examinations, and not all pathogens were tested for full antibiotic susceptibility according to our hospital’s standard procedures. The absence of data for some patients may impact the generalizability of our findings. Hence, future studies with larger sample sizes are warranted to address these limitations.

## 5. Conclusions

In summary, this retrospective multicenter cohort study has delineated the disparities in prevalence, characteristics, etiology, microbial profiles, and resistance patterns of orbital and preseptal cellulitis between patients before and during the COVID-19 pandemic. Based on our findings, it is recommended that clinicians implement enhanced monitoring of antibiotic resistance patterns, with particular attention paid to erythromycin, oxacillin, penicillin, and metronidazole. Heightened vigilance is warranted in patients with odontogenic origins of infection, which have shown increased prevalence during the pandemic period. Additionally, clinicians are advised to exercise caution regarding visual prognosis and to adapt antibiotic therapy to align with the shifting resistance patterns. Strategic and prudent antibiotic use, alongside regular updates on local microbial trends, may facilitate optimized treatment approaches in the post-pandemic context. Continued research and data collection are essential to maintain awareness of these evolving trends.

## Figures and Tables

**Figure 1 microorganisms-12-02262-f001:**
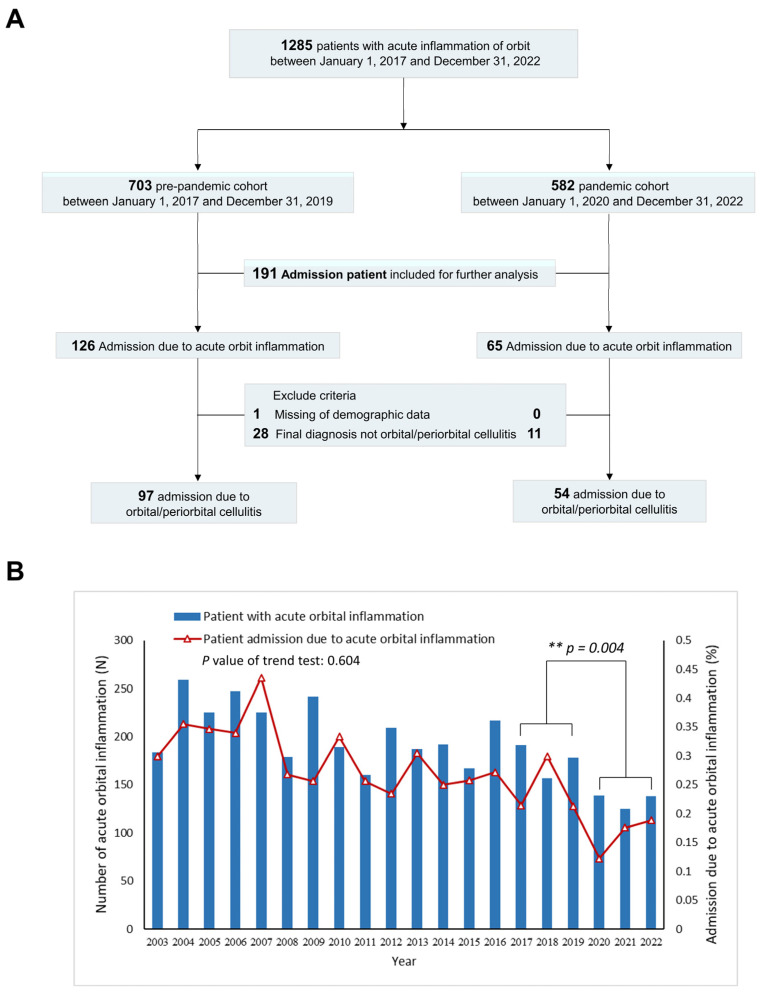
(**A**) Diagram illustrating the patient recruitment process (**B**) The prevalence of admission rates for orbital and preseptal cellulitis between 2003 and 2022. A significant decrease in admission rates was observed in the pandemic cohort compared to the pre-pandemic cohort (16.2% vs. 24.0%, *p* = 0.004). *p* values less than 0.01 are flagged with two asterisks (**) in the figure.

**Figure 2 microorganisms-12-02262-f002:**
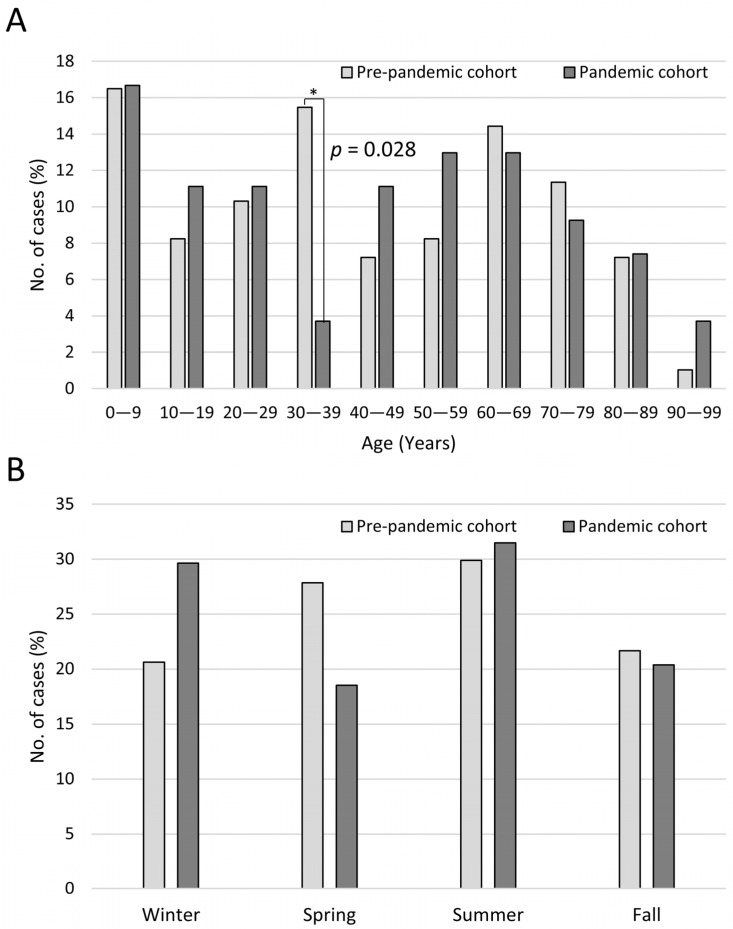
The prevalence of patients admitted for orbital and preseptal cellulitis. (**A**) Age and (**B**) seasonal distribution. *p* values less than 0.05 is flagged with one asterisk (*).

**Figure 3 microorganisms-12-02262-f003:**
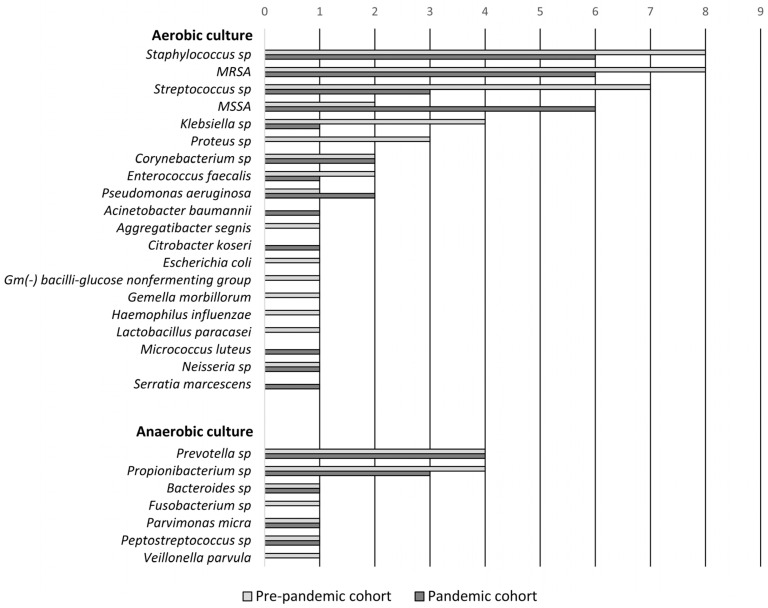
Pathogens isolated from localized noninvasive cultures. MRSA = methicillin-resistant *Staphylococcus aureus*; MSSA = methicillin-sensitive *Staphylococcus aureus*; sp = species.

**Figure 4 microorganisms-12-02262-f004:**
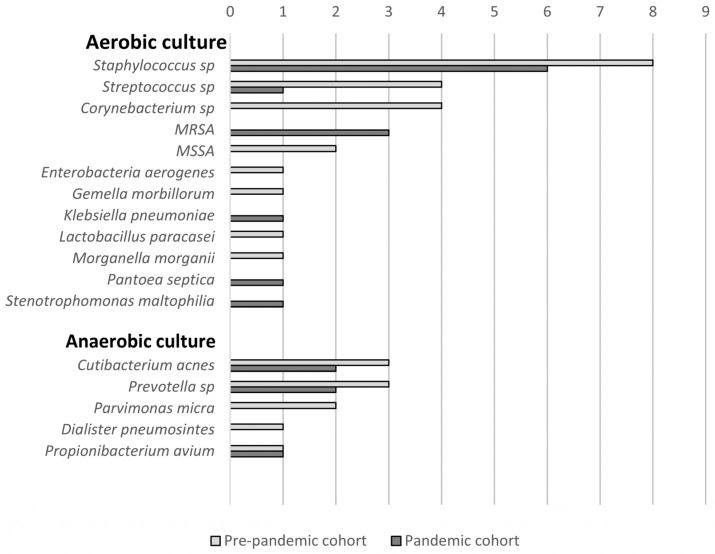
Pathogens isolated from surgical drainage samples. MRSA = methicillin-resistant *Staphylococcus aureus*; MSSA = methicillin-sensitive *Staphylococcus aureus*; sp = species.

**Table 1 microorganisms-12-02262-t001:** Characteristics of hospitalized patients.

	Pre-Pandemic Cohort	Pandemic Cohort	*p* Value
Basic Characteristics	(N = 97)	(N = 54)	
OD/OS/OU, n	47/46/4	23/26/5	0.407 ^α^
Age, mean ± sd (Years)	42.53 ± 26.78	43.54 ± 28.35	0.840 ^γ^
Sex, M/F	50/47	26/28	0.689 ^α^
BMI, mean ± sd	22.68 ± 5.21	22.15 ± 5.44	0.813 ^γ^
Disease diagnosis, n	Preseptal cellulitis: 15 Orbital cellulitis: 82	Preseptal cellulitis: 16 Orbital cellulitis: 38	**0.039** ^α^
Underlying disease			
Hypertension, n (%)	20 (20.6%)	17 (31.5%)	0.137 ^α^
Diabetes mellitus, n (%)	18 (18.6%)	8 (14.8%)	0.559 ^α^
Cancer, n (%)	8 (8.2%)	7 (13.0%)	0.353 ^α^
Other systemic diseases, n (%)	55 (56.7%)	30 (55.6%)	0.892 ^α^
Eyelid surgery history, n (%)	2 (2.1%)	0 (0%)	0.537 ^β^
Sinus surgery history, n (%)	9 (9.3%)	5 (9.3%)	0.997 ^α^
CT examination, n (%)	71 (73.2%)	39 (72.2%)	0.897 ^α^
Blood culture, n (%)	62 (63.9%)	38 (70.4%)	0.422 ^α^
Wound culture, n (%)	40 (41.2%)	31 (57.4%)	0.056 ^α^
Drainage surgery, n (%)	12 (12.4%)	10 (18.5%)	0.291 ^α^
Hospitalization days	11.54 ± 15.05	12.02 ± 17.59	0.239 ^γ^

^α^: Chi-square test; ^β^: Fisher’s exact test; ^γ^: Mann–Whitney U test; BMI: body mass index; CT: computed tomography; EOMs: extraocular muscles; F: female; M: male; OD: oculus dexter; OS: oculus sinister; OU: oculus uterque.

**Table 2 microorganisms-12-02262-t002:** Etiology.

	Pre-Pandemic Cohort	Pandemic Cohort	
	(N = 97)	(N = 54)	
Sinusitis, n (%)	36 (37.1%)	18 (33.3%)	0.642 ^α^
Skin infection, n (%)	12 (12.4%)	7 (13.0%)	0.916 ^α^
Hordeolum, n (%)	10 (10.3%)	1 (1.9%)	0.098 ^β^
Periocular trauma, n (%)	6 (6.2%)	4 (7.4%)	0.746 ^β^
Odontogenic, n (%)	2 (2.1%)	6 (11.1%)	0.025 ^β^
Periocular surgery, n (%)	6 (6.2%)	3 (5.6%)	1.000 ^β^
Dacryocystitis, n (%)	2 (2.1%)	4 (7.4%)	0.188 ^β^
Ocular origin, n (%)	3 (3.1%)	3 (5.6%)	0.667 ^β^
upper respiratory infection, n (%)	4 (4.1%)	1 (1.9%)	0.655 ^β^
Insect or animal bite, n (%)	3 (3.1%)	1 (1.9%)	1.000 ^β^
Sepsis, n (%)	3 (3.1%)	0 (0%)	0.553 ^β^
Periocular herpes infection, n (%)	1 (1.0%)	1 (1.9%)	1.000 ^β^
Unknown, n (%)	8 (8.2%)	2 (3.7%)	0.496 ^β^

^α^: Chi-square test; ^β^: Fisher’s exact test.

**Table 3 microorganisms-12-02262-t003:** Antibiotic Susceptibility Profiles of Pathogens Identified in Localized Noninvasive Cultures.

	Pre-Pandemic Cohort			Pandemic Cohort			*p* Value
	Sensitive (N)	Resistant (N)	Sensitivity Rate (%)	Sensitive (N)	Resistant (N)	Sensitivity Rate (%)	
**Aerobic**							
Amikacin	8	2	80.0%	6	0	100.0%	0.5 ^β^
Ampicillin	12	1	92.3%	3	1	75.0%	0.426 ^β^
Amoxicillin–clavulanic acid	3	2	60.0%	0	0	N/A	N/A
Ampicillin–sulbactam	2	0	100.0%	2	1	66.7%	1.000 ^β^
Ceftazidime	7	3	70.0%	5	1	83.3%	1.000 ^β^
Clindamycin	19	6	76.0%	19	3	86.4%	0.470 ^β^
Ciprofloxacin	8	1	88.9%	5	1	83.3%	1.000 ^β^
Cefoperazone–sulbactam	10	0	100.0%	4	1	80.0%	0.333 ^β^
Cefepime	3	1	75.0%	3	0	100.0%	1.000 ^β^
Ceftriaxone	12	3	80.0%	6	0	100.0%	0.526 ^β^
Cefuroxime	6	1	85.7%	1	1	50.0%	0.417 ^β^
Cefazolin	3	5	37.5%	1	1	50.0%	1.000 ^β^
Clarithromycin	1	0	100.0%	0	0	N/A	N/A
Clindamycin	1	1	50.0%	0	0	N/A	N/A
Colistin	2	0	100.0%	3	0	100.0%	N/A
Doripenem	2	0	100.0%	2	0	100.0%	N/A
Erythromycin	14	14	50.0%	14	8	63.6%	0.335 ^α^
Ertapenem	8	0	100.0%	2	0	100.0%	N/A
Gentamicin	8	3	72.7%	6	0	100.0%	0.515 ^β^
Levofloxacin	9	1	90.0%	3	0	100.0%	1.000 ^β^
Fusidic acid	10	0	100.0%	12	0	100.0%	N/A
Flomoxef	1	1	50.0%	0	0	N/A	N/A
Imipenem	2	0	100.0%	3	0	100.0%	N/A
Linezolid	10	0	100.0%	12	0	100.0%	N/A
Meropenem	2	0	100.0%	3	0	100.0%	N/A
Oxacillin	8	10	44.4%	11	8	57.9%	0.413 ^α^
Penicillin	13	15	46.4%	7	18	28.0%	0.167 ^α^
Piperacillin–tazobactam	7	0	100.0%	3	1	75.0%	0.364 ^β^
Sulfamethoxazole–trimethoprim	15	2	88.2%	16	2	88.9%	1.000 ^β^
Teicoplanin	27	0	100.0%	23	0	100.0%	N/A
Tetracycline	1	0	100.0%	0	1	0.0	1.000 ^β^
Tigecycline	12	0	100.0%	13	0	100.0%	N/A
Vancomycin	26	0	100.0%	23	0	100.0%	N/A
**Anaerobic**							
Ampicillin–sulbactam	11	1	91.7%	11	0	100.0%	1.000 ^β^
Clindamycin	11	2	84.6%	8	3	72.7%	0.630 ^β^
Metronidazole	9	4	69.2%	7	4	63.6%	1.000 ^β^
Penicillin	7	6	53.8%	8	3	72.7%	0.423 ^β^
Piperacillin	12	1	92.3%	10	1	90.9%	1.000 ^β^

^α^: Chi-square test; ^β^: Fisher’s exact test. N/A = not applicable.

**Table 4 microorganisms-12-02262-t004:** Antibiotic susceptibility profiles of pathogens identified in surgical drainage cultures.

	Pre-Pandemic Cohort			Pandemic Cohort			*p* Value
	Sensitive (N)	Resistant (N)	Sensitivity Rate (%)	Sensitive (N)	Resistant (N)	Sensitivity Rate (%)	
**Aerobic**							
Amikacin	2	0	100.0%	2	0	100.0%	N/A
Ampicillin	5	0	100.0%	1	0	100.0%	N/A
Ampicillin–sulbactam	1	1	50.0%	2	0	100.0%	1.000 ^α^
Ceftazidime	2	0	100.0%	2	0	100.0%	N/A
Clindamycin	10	3	76.9%	8	3	72.7%	1.000 ^α^
Ciprofloxacin	2	0	100.0%	2	0	100.0	N/A
Cefoperazone–sulbactam	2	0	100.0%	2	0	100.0%	N/A
Ceftriaxone	8	0	100.0%	3	1	75.0%	0.333 ^α^
Cefuroxime	0	2	0.0	2	0	100.0%	0.333 ^α^
Cefazolin	0	2	0.0	2	0	100.0%	0.333 ^α^
Clarithromycin	1	0	100.0%	0	0	N/A	N/A
Clindamycin	1	0	100.0%	0	0	N/A	N/A
Erythromycin	7	8	46.7%	4	8	33.3%	0.696 ^α^
Ertapenem	2	0	100.0%	2	0	100.0%	N/A
Gentamicin	2	0	100.0%	2	0	100.0%	N/A
Levofloxacin	2	0	100.0%	2	1	66.7%	1.000 ^α^
Fusidic acid	2	0	100.0%	4	0	100.0%	N/A
Linezolid	2	0	100.0%	4	0	100.0%	N/A
Moxifloxacin	0	0	N/A	1	0	100.0%	N/A
Oxacillin	5	5	50.0%	3	9	25.0%	0.378 ^α^
Penicillin	6	9	40.0%	1	11	8.3%	0.091 ^α^
Piperacillin–tazobactam	2	0	100.0%	2	0	100.0%	N/A
Sulfamethoxazole–trimethoprim	9	1	90.0%	10	2	83.3%	1.000 ^α^
Teicoplanin	15	0	100.0%	12	0	100.0%	N/A
Tetracycline	0	0	N/A	1	0	100.0%	N/A
Tigecycline	2	0	100.0%	3	0	100.0%	N/A
Vancomycin	15	0	100.0%	12	0	100.0%	N/A
**Anaerobic**							
Ampicillin–sulbactam	10	0	100.0%	7	0	100.0%	N/A
Clindamycin	10	0	100.0%	7	0	100.0%	N/A
Metronidazole	6	4	60.0%	2	5	28.6%	0.335 ^α^
Penicillin	7	3	70.0%	5	2	71.4%	1.000 ^α^
Piperacillin	10	0	100.0%	6	1	85.7%	0.412 ^α^

^α^: Fisher’s exact test. N/A = not applicable.

**Table 5 microorganisms-12-02262-t005:** Visual outcomes.

	Pre-Pandemic Cohort	Pandemic Cohort	*p* Value
Vision at admission, mean ± sd (LogMAR)	0.71 ± 0.85	0.76 ± 0.86	0.722 ^γ^
20/20–20/40, n (%)	32 (46.4%)	15 (45.5%)	0.930 ^α^
<20/40–5/200, n (%)	26 (37.7%)	12 (36.4%)	1.000 ^α^
<5/200, n (%)	11 (15.9%)	6 (18.2%)	0.782 ^β^
Vision at last follow-up, mean ± sd (LogMAR)	0.41 ± 0.68	0.74 ± 0.96	0.194 ^γ^
20/20–20/40, n (%)	17 (58.6%)	8 (53.3%)	0.737 ^α^
<20/40–5/200, n (%)	10 (34.5%)	5 (33.3%)	1.000 ^β^
<5/200-PL, n (%)	2 (6.8%)	2 (13.4%)	0.596 ^β^
Vision improved, n (%)	14 (53.8%)	9 (64.3%)	0.524 ^α^
Vision remained constant, n (%)	7 (26.9%)	3 (21.4%)	1.000 ^β^
Vision decrease, n (%)	5 (19.2%)	2 (14.3%)	1.000 ^β^
Vision improvement (LogMAR)	−0.29 ± 0.68	−0.19 ± 0.52	0.769 ^γ^

^α^: Chi-square test; ^β^: Fisher’s exact test; ^γ^: Mann–Whitney U test. PL = perception of light; SD: standard deviation.

## Data Availability

The raw data supporting the conclusions of this article will be made available by the authors on request.

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
