# Peer review of "Impact of the COVID-19 Pandemic on Microbial Profiles and Clinical Outcomes in Orbital and Preseptal Cellulitis"

_microorganisms, 2024, doi:10.3390/microorganisms12112262_

Round 1
Reviewer 1 Report
Comments and Suggestions for Authors
The authors should explain further why specific statistical methods (e.g., the Cochran-Armitage test) were used to analyze hospitalization trends. In addition, it would be helpful to justify the methodology used to assess microbial diversity and antibiotic resistance, discussing possible alternatives and their limitations.
Discussion of possible causes of the decrease in microbial diversity should be expanded. For example, explore how changing hygiene behaviors, mask use, or changes in access to healthcare during the pandemic might have influenced microbial selection. Compare these results to similar studies conducted in other geographic areas or clinical settings.
The authors should make more specific recommendations on how physicians can adjust treatments based on the study findings in the paper. For example, they could suggest closer monitoring of resistance to particular antibiotics, such as erythromycin, or prudent use of antibiotics in patients with orbital and preseptal cellulitis in the post-pandemic setting. Additionally, I recommend additional studies that can validate and expand these findings.
Reviewer 2 Report
Comments and Suggestions for Authors
Since the most common regimen used in both groups was amoxicillin-clavulanate, why was there no testing of susceptibility for amoxicillin -clavulanate in the pandemic cohort? (and also in pathogens identified after surgical drainage) ?
The visual acuity at admission was noted in 69 eyes in the pre-pandemic cohort, but only in 29 eyes at the last follow-up, while the calculation for improved/constant/decreased was done in 26 eyes. Why these discrepancies? Wasn't the VA noted at the patient's discharge from hospital?
(I do understand that this is not the main issue of this paper. If you have incomplete data about visual acuities, perhaps it would be better to acknowledge that and abstain to report the VA results).
in Table 1 : "Hypertnesion" (typo)
lines 276-277: "which limited us to thoroughly discuss the true pathogen" -unclear phrasing, please rephrase
I did not have access to any supplementary figures.
Reviewer 3 Report
Comments and Suggestions for Authors
This is a useful retrospective study of COVID-19 and ocular infections.
However, Sample and results:
There were 97 pre pandemic patients with non invasive cultures from 40 and surgical/invasive cultures form 12 (leaving 52 without culture results?) For the pandemic group of 54 there were 31 non invasive and 10 surgical/invasive cultures leaving 13 without cultures)
The key findings were:
“predisposing factors associated with preseptal and orbital cellulitis. In the pre-pandemic group, sinusitis, skin infection, and hordeolum emerged as the most prevalent predisposing factors, whereas in the pandemic group, sinusitis, skin infection, and odontogenic factors were predominant. Notably, the percentage of patients with odontogenic etiology was significantly higher in the pandemic group (11.1% compared to 2.1%, P = 0.025).”
The key pathogens were streptococcus MRSA and MSSA
[How is the generalisability of your study affected by the fact that you did not obtain cultures on a substantial number of patients? This significantly affects your conclusions]
Round 2
Reviewer 3 Report
Comments and Suggestions for Authors
Thank you for your clarification why you were not able to collect cultures from deep infections. Key information. Odd that the organisms did not float around to the cornea but perhaps the infections were encapsulated